# Nanobody-based VSR7 tracing shows clathrin-dependent TGN to Golgi recycling

Xiaoyu Shao [1,2], Hao Xu [2] & Peter Pimpl [2] ✉

Receptor-mediated transport of soluble proteins is nature's key to empowering eukaryotic cells to access a plethora of macromolecules, either by direct accumulation or as products from resulting biochemical pathways. The transport efficiency of these mechanisms results from the receptor's capability to capture, transport, and release ligands on the one hand and the cycling ability that allows for performing multiple rounds of ligand transport on the other. However, the plant VACUOLAR SORTING RECEPTOR (VSR) protein family is diverse, and their ligand-specificity and bidirectional trafficking routes and transport mechanisms remain highly controversial. Here we employ nanobody-epitope interaction-based molecular tools to assess the function of the VSR 7 in vivo. We demonstrate the specificity of the VSR7 for sequence-specific vacuolar sorting signals, and we trace its anterograde transport and retrograde recycling route. VSR7 localizes at the *cis*-Golgi apparatus at steady state conditions and transports ligands downstream to release them in the *trans*-Golgi network/early endosome (TGN/EE) before undergoing clathrin-dependent recycling from the TGN/EE back to the *cis*-Golgi.

A main distinction between the sorting of vacuolar proteins in plants and lysosomal proteins in mammals is the nature of the vacuolar/lysosomal sorting signal. In this regard, mammals utilize mannose-6-phosphate residues on *N*-linked oligosaccharyl chains that emerge upon posttranslational glycan modification in the *trans*-Golgi network[1], the last fenestrated cisternae of the Golgi apparatus. These signals are recognized by mannose-6-phosphate receptors[2] that operate between the TGN and the early endosome (EE) and the plasma membrane (PM) and the EE, respectively. In sharp contrast, plant vacuolar sorting signals are not based on glycosylation but are encoded in short amino acid sequences and are thus recognizable immediately upon protein synthesis and folding in the endoplasmic reticulum (ER)[3]. Albeit sequence-specific in their context, plant vacuolar sorting signals are diverse. Some sequence-specific vacuolar sorting signals (ssVSS) like the asparagine-proline-isoleucine-arginine-leucine (NPIRL) motif of the thiol protease aleurain from barley (*Hordeum vulgare*)[4] or the storage protein sporamin from sweet potato

(*Ipomoea batatas*)[5] locate in N-terminal propeptide of the respective protein and remain functional even if transplanted to the C-terminus[6]. However, the C-terminal vacuolar sorting signals (ctVSS) like the tetrapeptide alanine-phenylalanine-valine-tyrosine (AFVY) of the bean (*Phaseolus vulgaris*) storage protein phaseolin[7] strictly depends on their location of the utmost C-terminus[8]. Vacuolar targeting in plants is complicated because plant cells possess functionally different types of vacuoles for lysis (LV) and protein storage (PSV) that might originate independently[9,10] and may coexist[11] or transform into each other, dependent on tissue and physiological condition[12,13].

Plant vacuolar sorting receptors were discovered almost 30 years ago[14,15]. Soon after that, it became clear that two large gene families encode vacuolar sorting receptors in higher plants, the VACUOLAR SORTING RECEPTOR (VSR) family, which consists of seven members, termed VSR1-7 in *Arabidopsis*[16,17], and the RECEPTOR HOMOLOGY-TRANSMEMBRANE-RING-H2 (RMR) family, that consists of 6 family members in *Arabidopsis*[18-20]. In this regard, RMR proteins seem to

[1]Harbin Institute of Technology, Harbin, China. [2]Key Laboratory of Molecular Design for Plant Cell Factory of Guangdong Higher Education Institutes, Institute of Plant and Food Science, Department of Biology, School of Life Sciences, Southern University of Science and Technology (SUSTech), Shenzhen, Guangdong 518055, China. ✉e-mail: pimpl@sustech.edu.cn

facilitate protein transport to the PSV[20], while VSR sort to LVs and PSVs[21,22] The VSR family is grouped phylogenetically into three distinct classes: class I: VSR1 and VSR2; class II: VSR3 and VSR4; class III VSR5, VSR6, and VSR7)[16,17,23], (Supplementary Fig. 1). Most of our knowledge regarding VSR function results only from the analysis of the genetically redundant class I VSR1 and the class II VSRs 3 and 4, while information regarding the function of the class III VSRs 5, 6 and the most distant member, the VSR7, are scarce. In this study, we have analyzed the role of VSR7 in direct comparison to the VSR4 as a reference. We used fusion proteins of GFP- and α-synuclein-binding variable domains of heavy-chain antibodies from camelids termed nanobodies and epitope-tagged proteins as molecular tools to generate VSR7 sensors for assessing receptor-ligand interaction and compartment-specific tracing of its transport route. Our results identify the VSR7 as a *cis*-Golgi localizing VSR that transports ligands downstream to the *trans*-Golgi network/early endosome (TGN/EE) and recycles back to the *cis*-Golgi in a clathrin-dependent way.

## Results

### The VSR7 possesses ligand-binding ability

We have recently developed a strategy for assessing compartment-specific receptor-ligand interactions based on the nanobody-epitope interaction-triggered intermolecular assembly of sensor proteins in living cells[24]. To assess the function of the genetically most distant member of the VSR protein family, the VSR7 (Supplementary Fig. 1), we have employed a VSR7 sensor system that self-assembles from the GFP-binding nanobody (Nb$_G$)-tagged N-terminal luminal binding domain (LBD) of the VSR7, LBD7-Nb$_G$ as the sensing unit, and the GFP epitope-tagged, ER-localizing type I transmembrane anchor protein GFP-CNX in the lumen of the ER (Fig. 1a) for testing the interaction with a putative red fluorescent protein (RFP)-tagged ligand (Fig. 1b). Confocal laser-scanning microscopy (CLSM)-based analysis revealed that the coexpression of the ER-targeted VSR7 sensor GFP-CNX/LBD7-Nb$_G$ with the ssVSS-carrying reporter Aleu-RFP prevented the vacuolar delivery of the reporter and caused it's colocalization with the VSR7 sensor in the ER, instead (Fig. 1c). In control cells that lacked the Nb$_G$-tagged LBD7, Aleu-RFP did not accumulate in the ER but showed the typical vacuolar pattern of the reporter (Fig. 1d), suggesting that the observed ER accumulation occurred due to an interaction with the sensor in the ER. To obtain further direct evidence for VSR7 sensor-ligand interaction, we combined fluorescence lifetime imaging microscopy (FLIM) with conventional CLSM-based colocalization analysis. In this approach, FLIM allows the detection of occurring interactions between the GFP-based sensor and an RFP-based reporter protein as a quantifiable Förster resonance energy transfer (FRET)-induced reduction of the fluorescence lifetime of the energy-donating GFP. The resulting FLIM images, therefore, reveal the location-specific fluorescence lifetime of the GFP as a false color-encoded image, while the conventional CLSM 3-channel imaging shows the localization of the fluorescent proteins in the respective cell (Fig. 1e). We assessed ligand binding by recording the fluorescence lifetime of the GFP-based VSR7 sensor in the presence of the two commonly used vacuolar reporter proteins Aleu-RFP and RFP-AFVY as putative VSR7 ligands and the ER resident reporter protein RFP-HDEL as a non-ligand (Fig. 1e, Supplementary Fig. 2). The expression of Aleu-RFP caused a severe reduction of the GFP's fluorescence lifetime compared to the "donor only" control, which records the location-specific fluorescence lifetime of the GFP as a reference. The Aleu-RFP-caused reduction of fluorescence lifetime was almost as strong as the reduction that occurred due to the direct attachment of a red fluorescent fusion protein LBD7-RFP-Nb$_G$ by nanobody-epitope interaction, which served as a positive control to retrieve the strongest obtainable fluorescence lifetime reduction. Surprisingly, the ctVSS-carrying vacuolar reporter RFP-AFVY did not colocalize with the sensor in the ER, but exhibited the typical vacuolar pattern. The FLIM analysis revealed, that the

expression of the RFP-AFVY did not significantly change the VSR7 sensor's fluorescence lifetime, and the recorded values were comparable to those obtained when the ER-localizing non-ligand RFP-HDEL was expressed, instead. Together, this suggests that the LBD7 sensor specifically binds to the ssVSS-carrying vacuolar reporter Aleu-RFP but does not interact with the ctVSS of the vacuolar reporter RFP-AFVY in vivo.

The key to receptor-mediated sorting processes is binding, and release of ligands when sorting is complete. Ligand binding and release depend on the biochemical properties of a given compartment. Having shown that the lumen of the ER provides ligand binding conditions for the LBD of the VSR7, we assessed next the Golgi and the TGN/EE as the major waystations of the vacuolar transport route, whether they promote ligand binding or release. To achieve a Golgi-specific LBD7-ligand interaction analysis, we targeted the VSR7 sensing unit via nanobody-epitope interaction to the established GFP-epitope-tagged type II *cis*-Golgi-specific membrane protein anchor α-mannosidase 1, Man1-GFP as previously reported[24] (Fig. 2). Coexpression of the VSR7 sensor components with the ligand Aleu-RFP and a neutral Golgi marker Man1-cyan fluorescent protein (CFP) triggered an almost perfect colocalization of the ligand Aleu-RFP with the VSR7 sensor in the Golgi (Fig. 2a). This colocalization strictly depends on the coexpression of the receptor domain and is not seen in controls lacking a luminal binding domain[24] (Fig. 2b), in which the Aleu-RFP colocalises with the MVB/LE marker CFP-BP80, a fluorescent LBD-deprived derivative of the binding protein of 80 kDa (BP80), instead (Supplementary Fig. 3). In sharp contrast, TGN/EE targeting of the VSR7 sensor via the type II TGN/EE-specific fluorescent membrane anchor syntaxin of plants 61, SYP61-GFP, did not cause an evident colocalization of Aleu-RFP neither with the VSR7 sensor, or the Golgi marker (Fig. 2c), and appeared to be identical to controls lacking the LBD7-Nb$_G$ (Fig. 2d), indicating that the TGN/EE could be the release compartment of the VSR7, as was previously reported for the VSR4[24]. However, due to the strong vacuolar background signals of the Aleu-RFP, it is difficult to quantitatively judge whether or not small amounts of the ligand interacted with the LBD7 sensor in this crucial compartment. To overcome this obstacle, we envisaged performing a quantitative FLIM analysis in which the readout is not perturbed by even strong fluorescent signals from compartments, other than the sensor location. We targeted the LBD7-Nb$_G$ sensing unit via the compartment-specific membrane anchor SYP61-GFP to assemble VSR7 sensors in the TGN/EE, and assessed the fluorescence lifetime changes of the GFP-based VSR7 sensor in the coexpression with either Aleu-RFP or RFP-AFVY (Fig. 2e). Althougu expressed to high levels, neither the ligand Aleu-RFP, nor RFP-AFVY altered significantly the fluorescence life time of the sensor compared to the dual-color sensor SYP61-GFP/LBD7-RFP-Nb$_G$ as positive control. This shows, that the in the early secretory pathway observed interaction between the VSR7 sensor and Aleu-RFP (Fig. 1e) did not occur in the TGN/EE, and thus identifies the TGN/EE as the first non-binding compartment of the vacuolar transport route. We, therefore, speculated, that the TGN/EE could be the release compartment of the VSR7-mediated transport route. VSR-ligand interaction is pH dependent, with ligand binding at neutral pH and release of ligands at acidic pH[14]. Since the TGN/EE is the most acidic compartment of the vacuolar route[25], we speculated that the lack of sensor-ligand interaction could be due to the low pH of the TGN/EE. To test for this, we employed the drug concanamycin A (Conc A), a specific inhibitor of vacuolar-type H$^+$-ATPases (V-ATPases) that was shown to affect the TGN/EE-localizing vacuolar-type H$^+$-ATPase subunit a1 (VHA-a1)[26] thereby inhibiting the acidification of the TGN/EE[25], and performed a FLIM-based ligand interaction analysis (Fig. 2e). The drug strongly reduced the vacuolar pattern of Aleu-RFP and AFVY in the localization analysis and increased the fluorescence lifetime of the donor-only control to values similar to those recorded in the FLIM analysis with the ER-targeted GFP-CNX-based sensor before (compare Fig. 2e (V) to Fig. 1e (I)), suggesting the

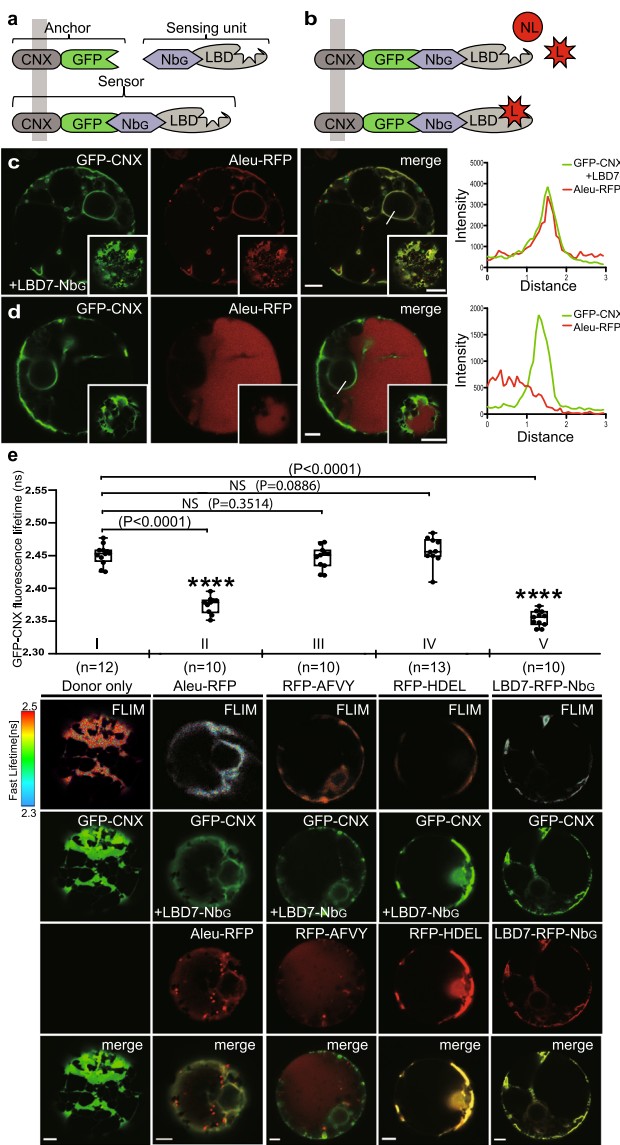

**Fig. 1 | The VSR7 sensor binds sequence-specific vacuolar sorting signals. a** ER-targeting and self-assembly of VSR7-sensors from a compartment-specific epitope-tagged membrane anchor and a nanobody-tagged sensing unit by nanobody-epitope interaction. The C/N-terminus of respective domains within a fusion protein is indicated in the nomenclature, rather than the illustration. **b** Colocalization-based VSR7 sensor-ligand interaction analysis with ligands (L) and non-ligands (NL). **c**–**e** CLSM analysis of VSR7 sensor-ligand interaction in tobacco mesophyll protoplasts, source data are provided as a Source Data file. **c** The assembly of VSR7 sensors from the coexpressed GFP-CNX with LBD7-Nb$_G$ (green) traps the vacuolar reporter Aleu-RFP (red) in the ER, as judged by overlapping signals in the nuclear envelope and overlapping peaks in the line intensity plot. **d** Control cells that coexpress only the GFP-CNX anchor (green) and Aleu-RFP (red) but lack the sensing unit LBD7-Nb$_G$ show no overlapping signals in the ER. **c, d** Experiments were repeated thrice with similar results, representative images are shown. **e** Quantitative FLIM assessment of VSR7 sensor-ligand interactions, showing (I) Energy donor (GFP-CNX) only, or the assembled VSR7 sensor GFP-CNX/LBD7-Nb$_G$ with (II) Aleu-RFP, (III) RFP-AFVY, (IV) RFP-HDEL or (V) the GFP-RFP dual-color sensor GFP-CNX/LBD7-RFP-Nb$_G$. Fluorescence lifetimes are given as GFP-CNX fluorescence lifetime in ns. All values are given in a box and whisker plot from min to max, with the box showing the interquartile range of the middle 50%; the median is indicated by a line. Only Aleu-RFP triggers the reduction of the VSR7 sensor's fluorescence lifetime, while RFP-AFVY and RFP-HDEL did not. Significance was calculated using one-way ANOVA, followed by student's t-test (****$P < 0.0001$ compared with every other group; NS, not significant). Sample sizes (n) and P values are given in the Figure. Images in the rows below the chart show false color fluorescence lifetime analysis (FLIM), and the respective localization analysis of the indicated proteins (green channel, GFP-CNX; red channels, Aleu-RFP, RFP-AFVY, RFP-HDEL, and LBD7-RFP-Nb$_G$, and the respective merges). Scale bars = 5 μm. Experiments were repeated twice with similar results; representative images are shown.

change of the compartmental pH perturbed the vacuolar transport. At these conditions, Aleu-RFP strongly reduced the fluorescence lifetime of the VSR7 sensor to levels comparable to the dual-color sensor SYP61-GFP/LBD7-RFP-Nb$_G$, but RFP-AFVY had no significant influence on the fluorescence lifetime of the VSR7 sensor. Together, this suggests that the VSR7-ligand interaction is pH dependent, with ligand binding at neutral pH in the early secretory pathway and release at acidic pH in the TGN/EE.

### The VSR7 is a *cis*-Golgi-localizing VSR that cycles between the *cis*-Golgi and the TGN/EE

To elucidate VSR7 function, we performed a localization analysis using a fluorescent, HA-tagged full-length receptor, GFP-VSR7 (Fig. 3), in coexpression with the TGN/EE-localizing fluorescent full-length VSR4, RFP-VSR4 (Supplementary Fig. 4) and fluorescent markers for the TGN/EE (SYP61-RFP), the MVB/LE (RFP-BP80), the *trans*-Golgi (sialyl transferase (ST)-RFP), and the *cis*-Golgi (Man1-RFP) (Fig. 3a–e). Surprisingly, at steady state conditions, GFP-VSR7 colocalized only with the *cis*-Golgi marker Man1-RFP, and no overlap of signals was recorded in case of coexpression with any of the other compartmental markers or the VSR4 (see Supplementary Fig. 5 for quantification). This suggests that the VSR7 is a *cis*-Golgi-localizing vacuolar sorting receptor.

Since our VSR7 sensor-based ligand binding analysis identified the *cis*-Golgi as a compartment that promotes ligand-binding, we tested next the *cis*-Golgi-localizing full-length VSR7 for its ligand-specificity regarding the two types of VSS (Fig. 3f–i). Coexpression of GFP-VSR7 together with the ssVSS-carrying ligand Aleu-RFP or the ctVSS-carrying non-ligand RFP-AFVY together with *cis*-Golgi or TGN/EE marker, respectively, revealed an overlap of signals from GFP-VSR7 and Aleu-RFP (Fig. 3f, g) but not RFP-AFVY (Fig. 3h, i), demonstrating that the full-length VSR binds specifically to the ssVSS but not to the ctVSS in vivo. Interestingly, irrespective of the coexpression with the ligand Aleu-RFP or the non-ligand RFP-AFVY, all GFP-VSR7 signals were found to localize only in the *cis*-Golgi, the identified ligand binding compartment (Fig. 3f, h) but none of the receptor signals were found to localize in the TGN/EE, the identified ligand-release compartment (Fig. 3g, i). This raised the question of whether and how a *cis*-Golgi-localizing VSR can mediate the transportation of ligands in the vacuolar route. Based on our ligand-binding analysis, we hypothesized that the *cis*-Golgi-localizing VSR7 could travel downstream to the TGN/EE for ligand release.

We have previously implemented nanobody/epitope-tagged fusion proteins to demonstrate the retrograde upstream recycling of the TGN/EE-localizing VSR4 to the *cis*-Golgi[27]. We wanted to take further advantage of the specificity and sensitivity of nanobody-epitope interaction-triggered intermolecular assembly reactions to test our hypothesis regarding putative downstream trafficking of the *cis*-Golgi-localizing VSR7 to the TGN/EE. For this, we envisaged using Nb$_G$-tagged receptors together with GFP epitope-labeled TGN/EE-localizing membrane anchor proteins, which would catch and trap transiting receptors immediately upon their arrival in the TGN/EE via nanobody-epitope interaction (Fig. 4a). However, the realization of such an approach in vivo is complicated by the fact that translational nanobody- and epitope-tagged fusion proteins cannot simply be coexpressed for this purpose because they will bind to each other already in the ER, immediately after synthesis and folding is completed. In this case, the proteins would travel alongside, and it would be impossible

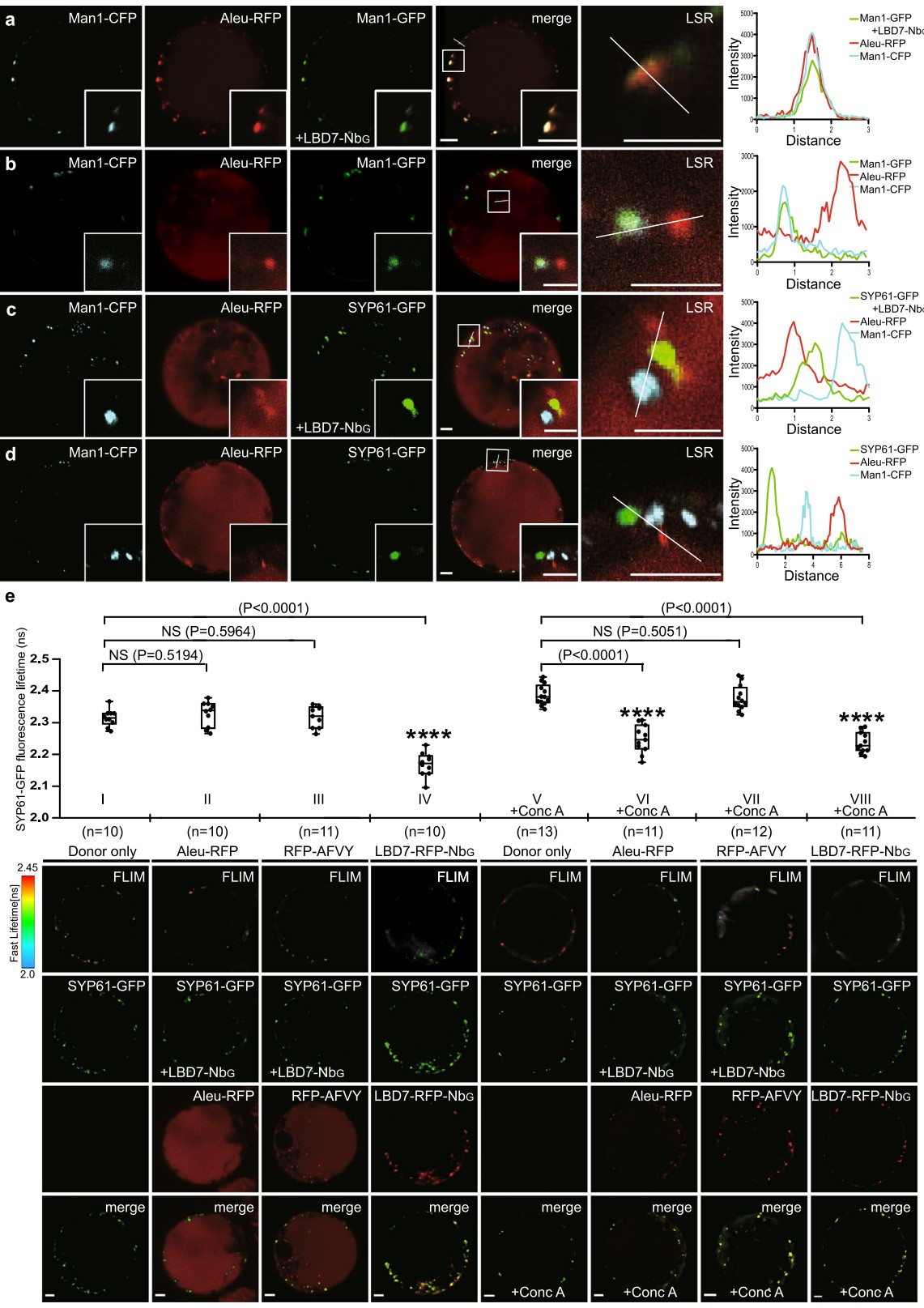

to set up a trap for incoming receptors at a downstream location. To overcome these constraints, we outsourced the synthesis of the required GFP-epitope and opted for an early-endosomal-location-specific posttranslational GFP-labelling of the TGN/EE-localizing membrane anchor, instead of using a translational GFP-fusion protein. To achieve this, we employed an additional nanobody-epitope pair, the α-synuclein binding nanobody (Nb$_S$), and its 23 amino acid-long

epitope, SYN, to drive the posttranslational attachment of the GFP-epitope as an SYN-tagged GFP protein, GFP-SYN[27], to an Nb$_S$-tagged TGN/EE membrane anchor. This dual-epitope protein GFP-SYN is produced as a secretory protein (secGFP-SYN) by a separate population of protoplasts, is recovered from their culture medium, and is used in a second step for the posttranslational labeling in endocytic uptake experiments with protoplasts, expressing the respective

**Fig. 2 | Compartment-specific targeting of VSR7 sensors reveals ligand binding in the Golgi and ligand release in the TGN/EE. a-d** Coexpression of **a** Golgi-targeted sensor components (Man1-GFP and LBD7-Nb$_G$ (green)) or **c** TGN/EE-targeted sensor components (SYP-61-GFP and LBD7-Nb$_G$ (green))with the ligand Aleu-RFP (red) and the Golgi marker Man1-CFP (blue) reveals only colocalization of the ligand with the Golgi-targeted sensor but not with TGN/EE-targeted sensors. **b, d** Control cells lacking LBD7-Nb$_G$ show no colocalization between Aleu-RFP and the respective markers/anchors. Insets are magnifications of the region indicated by a white square in the merged image. Line scan intensity plots visualize signal colocalization/distribution. LSR, line scan region. **a–d** Experiments were repeated thrice with similar results; representative images are shown. **e** TGN/EE-specific FLIM assessment of sensor-ligand interactions in absence/presence of 20 µM con-canamycin A ( + Conc A) for 12 h, showing (I) Energy donor (SYP61-GFP) only, or assembled VSR7 sensors (SYP61-GFP/LBD7-Nb$_G$) with (II) Aleu-RFP, (III) RFP-AFVY, and (IV) the GFP-RFP dual-color sensor SYP61-GFP/LBD7-RFP-Nb$_G$ (reference for maximum lifetime reduction) without Conc A, and V) Energy donor (SYP61-GFP)

only, the assembled VSR7 sensor SYP61-GFP/LBD7-Nb$_G$ with (VI) Aleu-RFP, (VII) RFP-AFVY, and (VIII) the GFP-RFP dual-color sensor SYP61-GFP/LBD7-RFP-Nb$_G$ with Conc A. Fluorescence lifetimes are given as SYP61-GFP fluorescence lifetime in ns. The box-whisker plot shows all values from min to max, with the box showing the interquartile range of the middle 50%; the median is indicated by a line. Conc A increases the pH in the TGN[25], causing longer lifetimes, thus necessitating another donor-only recording[54]. Aleu-RFP strongly reduces the sensor's fluorescence life-time in the presence of the drug (VI), comparable to the positive control (VIII), revealing that reduced TGN/EE acidification prevents ligand release. Significance was calculated using one-way ANOVA, followed by student's t-test (****$P < 0.0001$ compared with every other group; NS, not significant). Sample sizes (n) refer to cell numbers and are given together with $P$ values in the Figure. Rows below show false color FLIM images and the respective localization analysis (green/red/merged channels). Scale bars = 5 µm. Experiments were repeated twice with similar results; representative images are shown. Source data for **a**–**e** are provided as a Source Data file.

Nb$_G$/Nb$_S$-tagged receptor/anchor proteins[24,27]. In this scenario, the SYN epitope of the endocytosed dual-epitope protein secGFP-SYN would posttranslationally label the Nb$_S$-tagged cyan fluorescent TGN/EE marker SYP61-CFP-Nb$_S$, while the GFP epitope would serve as the bait for trapping the arriving Nb$_G$-tagged red fluorescent VSRs, Nb$_G$-RFP-VSR7 (Fig. 4aII, see Supplementary Fig. 6 for quantification). For the successful application of this strategy, it was important to ensure that the Nb$_G$-tagging of the red fluorescent VSR7 does not alter the *cis*-Golgi localization of the receptor (Fig. 4b). Most important however was to demonstrate that the endocytosed dual-epitope linker secGFP-SYN follows the default route to the vacuole, but does not reach the *cis*-Golgi. For this we performed endocytic uptake assays showing that the endocytosed secGFP-SYN does not colocalize with the *cis*-Golgi marker Man1-CFP (Fig. 4c), or even with the Nb$_G$-tagged *cis*-Golgi marker Man1-RFP-Nb$_G$ (Fig. 4d).

Of similar importance was to ensure that the coexpression of the two nanobody-tagged proteins Nb$_G$-RFP-VSR7 and SYP61-CFP-Nb$_S$, does not result in any colocalization of the fusion proteins in the absence of the dual-epitope linker protein due to unforeseeable unspecific inter-actions (Fig. 4e). In sharp contrast to the above controls, incubation of Nb$_G$-RFP-VSR7 and SYP61-CFP-Nb$_S$-coexpressing cells with the protoplast-secreted secGFP-SYN caused an almost complete overlap of the signals from the VSR7, the TGN/EE anchor and the dual epitope linker secGFP-SYN (Fig. 4f). Such colocalizations between the VSR7 and the TGN/EE anchor were never observed in the absence of either the linking secGFP-SYN (e), or if the conventional TGN/EE marker SYP61-CFP, which lacks the Nb$_S$, is used, instead (Fig. 4g). This demonstrates that the *cis*-Golgi-localizing Nb$_G$-RFP-VSR7 does indeed travel down-stream of its steady-state location, and reaches the TGN/EE, where it is linked to the posttranslationally secGFP-SYN-labelled TGN/EE anchor via the Nb$_G$-GFP nanobody-epitope interaction. Interestingly, incuba-tion of cells coexpressing the Nb$_G$-RFP-VSR7 and the TGN/EE marker SYP61-CFP with the secGFP-SYN results in a clear overlap of signals of only the Nb$_G$-RFP-VSR7 and the endocytosed secGFP-SYN linker, while no colocalization of these signals was detected with the SYP61-CFP (Fig. 4g). This colocalization, however, shows that the Nb$_G$-tagged VSR7 came in contact with the endocytosed secGFP-SYN in the TGN/EE and consequently, became posttranslationally labeled. This furthermore demonstrates that the secGFP-SYN-labeled Nb$_G$-RFP-VSR7 did indeed transit the TGN/EE, and we speculated whether the non-TGN/EE-loca-lizing secGFP-SYN-labeled VSR7 are receptors that have already recy-cled from the TGN/EE. To test for this, we subjected Nb$_G$-RFP-VSR7 and *cis*-Golgi marker Man1-CFP-expressing cells to endocytic uptake assays with secGFP-SYN (Fig. 4aIII, see Supplementary Fig. 6 for quantification) and found that signals from the posttranslationally secGFP-SYN-labeled Nb$_G$-RFP-VSR7 colocalize of with the *cis*-Golgi marker (Fig. 4h). Toge-ther, this demonstrates that the *cis*-Golgi-localizing VSR7 already experienced downstream traveling, TGN/EE transit and upstream

recycling to the *cis*-Golgi. The demonstrated TGN/EE transit of the VSR7 is also supported by its detection in fractions of immunoisolated SYP61-CFP or vacuolar-type H$^+$-ATPase subunit a1 (VHA-a1)-GFP compartments in proteomic analyses[28,29] even though they are non-discriminative for proteins that have multiple locations and proteins that reside exclu-sively in these compartments[29].

## The recycling of the *cis*-Golgi-localizing VSR7 depends on clathrin

Our data show that the VSR7 localizes at steady-state conditions in the *cis*-Golgi, which differs from the TGN/EE localization of the best-characterized member of the VSR protein family, the VSR4 (Fig. 3a and Supplementary Fig. 4). We speculated that the differences in the localization could be due to differences in their Golgi trafficking. Golgi trafficking might involve coat protein I (COPI)-coated vesicles that are formed after the activation and membrane recruitment of the COPI-specific GTPase ADP-ribosylation factor 1, ARF1, which in turn, recruits the heptameric cargo-recognizing coatomer complex. To test for dif-ferences in the transport of the two receptors, we coexpressed them with the GTP-locked dominant-negative ARF1 mutant, ARF1M[30], and the *cis*-Golgi marker or the TGN/EE marker, respectively (Fig. 5a–c, Supplementary Fig. 8, see Supplementary Fig. 7 for quantification). The coexpression of ARF1M altered the colocalization of the RFP-VSR7 with Man1-CFP and caused its colocalization with GFP-VSR4, instead (Fig. 5a), which was never seen in the absence of the ARF1M (Fig. 5b, Supplementary Fig. 8), suggesting, that the VSR7 has lost its *cis*-Golgi localization. However, the expression of the ARF1M did not affect the TGN/EE localization of the VSR4 (Fig. 5c, compared to Supplementary Fig. 4), demonstrating that the mutant did not affect the anterograde transport across the Golgi stack and beyond to the TGN/EE. Therefore, we speculated that the ARF1M-caused TGN/EE colocalization of the VSR7 with the VSR4 hints at an ARF1-dependent VSR7 recycling mechanism from the TGN/EE, ultimately leading to the *cis*-Golgi loca-lization of the VSR7 at steady-state conditions.

ARF1 has been shown to localize at the TGN/EE in addition to the Golgi stack, while the localization of the coatomer complex seems to be restricted to the Golgi stack. Interestingly, ARF1 also plays a crucial role in the recruitment and activation of tetrameric clathrin adaptor complexes that mediate the cargo selection and formation of clathrin-coated vesicles (CCVs)[31], the most prominent type of transport vesi-cles, found to bud at the TGN/EE. Based on this, we speculated that the VSR7 recycles after the release of the ligand in the TGN/EE via a clathrin-dependent transport back to the *cis*-Golgi.

To test this hypothesis, we assessed the localization of the VSR7 regarding its clathrin dependency by coexpressing the dominant-negative mutant of the clathrin heavy chain, the clathrin hub[32], to inhibit clathrin-mediated trafficking (Fig. 5d–h, see Supplementary Fig. 7 for quantification). Coexpression of the fluorescent hub

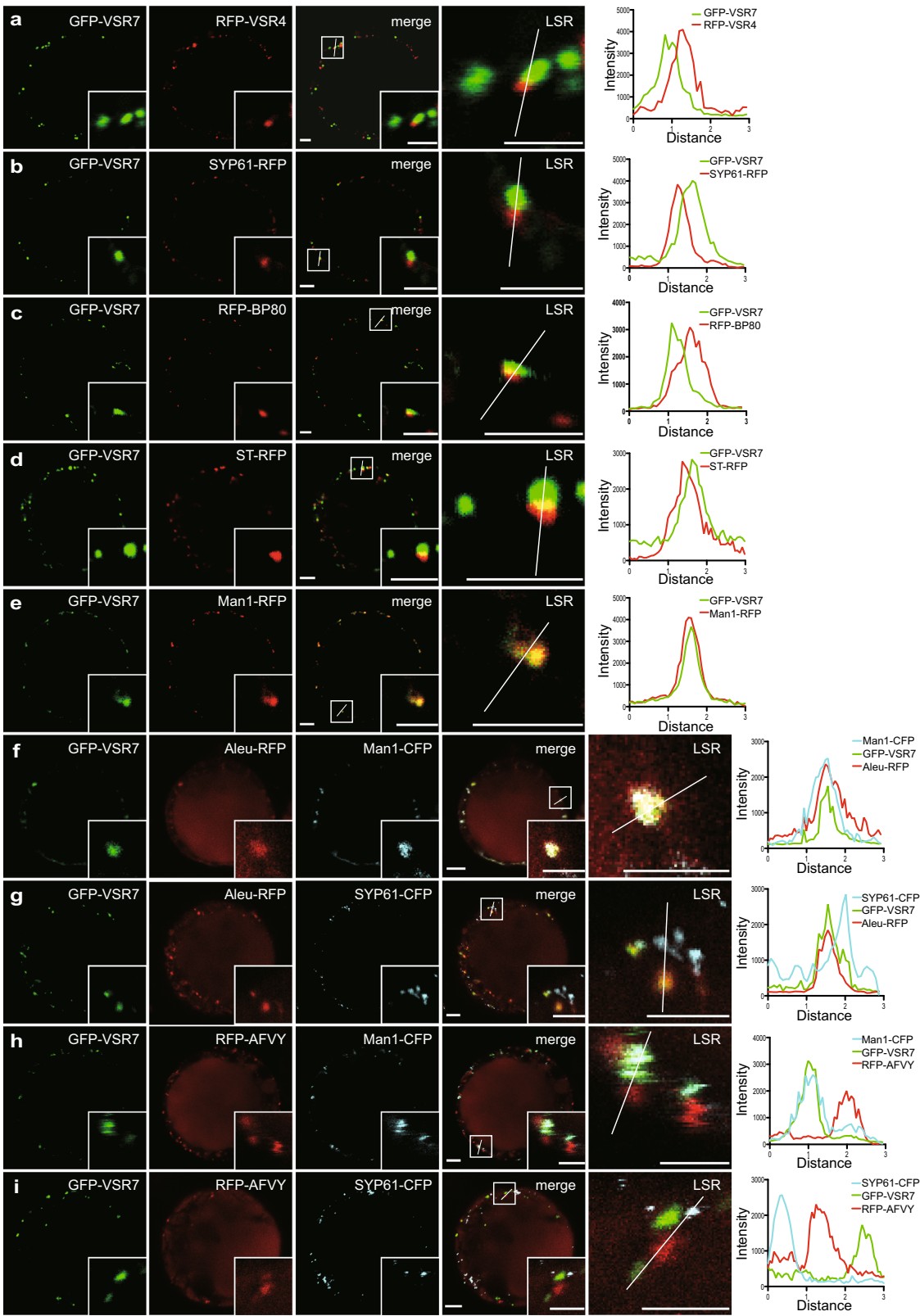

fragment, GFP-Hub, with the RFP-VSR7 and SYP61-CFP trapped the VSR7 in the TGN/EE, as judged by the colocalization of RFP-VSR7 and SYP61-CFP signals (Fig. 5d), similar to the signals obtained by the coexpression of the ARF1M (compare to overlapping red/blue peaks in the line intensity plots in Fig. 5c). Remarkably, some of the RFP-VSR7 signals did not colocalize with the TGN/EE marker. Therefore, we tested whether the Hub-caused recycling inhibition could have resulted in

the downstream progression of RFP-VSR7 to the MVB/LE, but this was not the case and no colocalization between the RFP-VSR7 and the MVB/LE marker CFP-BP80 was evident (Fig. 5e). However, we speculated, that the inhibition of the receptor recycling might not have been completed under these conditions. To test this hypothesis, we increased the expression time. After 48 h incubation, we found largely overlapping signals between the RFP-VSR7 and the CFP-BP80,

**Fig. 3 | The full-length VSR7 localizes to the *cis*-Golgi and exhibits ligand specificity. a–e** CLSM localization analysis of the fluorescent full-length VSR7, GFP-VSR7 (green) based on coexpression with compartment-specific marker proteins (red): **a** TGN/EE-localizing VSR4, **b** TGN/EE marker SYP61-RFP, **c** MVB/LE marker RFP-BP80, **d** *trans*-Golgi marker ST-RFP, **e** the *cis*-Golgi marker Man1-RFP reveals colocalization with the *cis*-Golgi marker Man1-RFP. Despite the occurrence of obviously non-colocalizing signals, signals in close proximity were analyzed by line scan analysis for verification. **f-i** Colocalization-based ligand binding analysis of GFP-VSR7 (green) with either Aleu-RFP (red) or RFP-AFVY (red) in the *cis*-Golgi and the TGN/EE. Coexpression of GFP-VSR7 and Aleu-RFP with **f** the *cis*-Golgi marker

Man1-CFP (blue) or **g** the TGN/EE marker SYP61-CFP (blue) showing VSR7-dependent Aleu-RFP accumulation in the *cis*-Golgi. Coexpression of GFP-VSR7 and RFP-AFVY with **h** Man1-CFP or **i** SYP61-CFP showing no VSR7-dependent RFP-AFVY accumulation of this non-ligand. The colocalization/distribution of fluorescent proteins are visualized by overlapping/non-overlapping peaks in the line intensity plots. Insets in **a**–**i** are magnifications of the region indicated by a white square in the merged image. LSR, line scan region of the line intensity analysis. Scale bars = 5 μm. Experiments **a**–**i** were repeated at least thrice with similar results; representative images are shown. Source data for **a**–**i** are provided as a Source Data file.

demonstrating the downstream progression of the VSR due to the inhibited recycling (Fig. 5f). Strikingly, the GFP-Hub did not alter the TGN/EE localization of the RFP-VSR4, and even after 48 h expression, the RFP-VSR4 did not reach the MVB/LE at detectable amounts (Fig. 5g) and remained in the TGN/EE (Fig. 5h). Together, this suggests, that the VSR7 recycles via a clathrin-dependent transport mechanism from the TGN/EE to the *cis*-Golgi, while the VSR4 does not join this trip.

## Discussion

Despite the apparent differences in morphology and organization of the intracellular compartments of eukaryotic cells, receptor-mediated transport of soluble proteins follows a common principle: the sorting receptors capture the soluble proteins via their ligand binding domain (LBD) in the compartmental lumen, thereby linking the soluble cargo protein via their cytosolic domain to the membrane trafficking machinery, which in turn fulfills the sorting and targeting of the receptor-ligand complex. As far as the ligand is concerned, receptor-mediated transport ends with the dissociation of the receptor-ligand complex in the chemically different environment of the target compartment. The receptor, however, undergoes retrograde recycling back to the donor compartment and performs further transport rounds. This principle is conserved among eukaryotes, but the implementation seems to vary to take into account the morphological peculiarities of the respective system.

Despite the different nature of vacuolar/lysosomal sorting signals, MPRs and VSRs exhibit pH-dependent receptor-ligand interaction[14,33,34], with ligand binding occurring at neutral to slightly acidic pH while ligand release occurs at acidic or alkaline pH. In plants, neutral to slightly alkaline conditions are found in the ER and the Golgi stack, while the *trans*-Golgi network, which is also the early endosome (TGN/EE)[26,35], is the most acidic compartment of the vacuolar transport route[25,36]. This supports the findings from the compartment-specific receptor-ligand interaction analysis, showing that plant receptors bind ligands in the early secretory pathway[3,24,37] and release them already in the TGN/EE[24], rather than in the multivesicular body, which is the late endosome (MVB/LE)[38], as was initially proposed[14,39], before they recycle upstream to the *cis*-Golgi for further rounds of transport[27]. The post-TGN/EE trafficking of the released ligands to the vacuole occurs then, together with all of the endocytosed soluble material independent of VSRs[24,40] by the TGN/EE maturation-based formation of MVBs/LEs[41], which ultimately fuse with the tonoplast[24,41].

Our direct comparison between VSR7 and VSR4 shows, that both receptors bind the ssVSS ligand in the ER and the Golgi, but not in the TGN/EE, suggesting that the VSR7 transports ligands from the early secretory pathway to the TGN/EE, as was previously demonstrated for the VSR4[24]. However, both receptors exhibit differential locations at steady-state conditions, with the VSR4 localizing to the TGN/EE and the VSR7 localizing at the *cis*-Golgi. Protein transport across the Golgi stack is not fully understood. It has been suggested that anterograde transport occurs via a cisternal maturation process that is driven by arriving material at the *cis*-face, while the Golgi-residing enzymes are constantly retrieved via COPI vesicles to maintain cisternal functionality, thereby resulting in a constant forward movement of the cisternae, which ultimately develop into a TGN that detaches from the stack thereby becoming a Golgi-independent TGN/EE[42,43]. In such a

scenario, the anterograde movement of membrane proteins like the VSRs would occur at the same speed. Therefore, it is plausible to assume that the differential location of the VSRs might be due to differences in their recycling from the TGN/EE after ligand release. In mammals, MPRs recycle from the EE to the TGN via the retromer complex[44], which was suggested to form tubular carriers[45]. In plants, recycling from the TGN/EE is controversial. However, subunits of the retromer complex have also been detected at the TGN/EE[29,46], biochemical evidence for a VSR1-retromer interaction was presented[47], and retromer subunit VPS29 knockdown mutants showed perturbed VSR1 recycling[48], suggesting that such a tubular carrier could indeed facilitate recycling of the genetically redundant VSRs. On the other, MPRs and VSRs alike possess characteristic tyrosine-based sorting motifs, which are recognized by the μ-adaptin subunit of tetrameric clathrin adapter protein complexes[49], to facilitate the sorting of membrane proteins into clathrin-coated vesicles. Together with our observation that the inhibition of clathrin-mediated trafficking perturbed only the recycling of the VSR7, but not the VSR4, it seems that these receptors use different recycling mechanisms that might differ in the time it takes to form and move a carrier for efficient cargo export from the TGN/EE. In this regard it is tempting to speculate that the loading and formation of a large tubular retromer-coated carrier could take longer than the formation of a smaller, spherical, clathrin-coated vesicle. However, this could account for a longer TGN/EE transit time and, thus, extended visibility of the departing VSR4 compared to the VSR7 (Fig. 5i). The TGN/EE is characterized by budding clathrin-coated vesicles (CCVs) and it was recently demonstrated that it possesses noticeably different subdomains to facilitate differential cargo export[42]. In this regard, it was suggested that the clathrin adaptor complex 1 (AP1) facilitates transport to the PM. Interestingly, AP1 seems to mediate multiple transport routes in mammals and yeats[50], including the recycling from a late-stage TGN to an early-stage[51]. Our data regarding the compartment-specific receptor-ligand interactions and the trafficking routes of the VSRs VSR4 and VSR7 point to a sorting and transport mechanism for the VSRs, that differs from the basic concept of receptor-mediated transport. Ligand transport is assumed to occur via a vesicle shuttle that connects spatially separated but persistent compartments, which provide ligand-binding or ligand-release conditions. However, such a model seems not apply to the morphological situation in plants. It is, therefore, tempting to speculate that plant VSRs mediate transport not between spatially separate persistent compartments but temporally distinct units. The TGN/EE as the target compartment ultimately emerges from the initial starting compartment in a maturation-based process. It appears as if the sorting function of a VSR is the ligand binding at a neutral pH in the *cis*-Golgi cisternae and thus its immobilization to prevent its secretory loss throughout the cisternae's maturation into a VHA-a1-acidified Golgi-independent TGN with EE functionality. This is in agreement with our demonstration of the pH-dependent ligand release in the TGN/EE, and also supports a recently suggested transport model in which the VSRs segregate vacuolar and secretory cargo in domains within the maturing TGN compartment[52]. However, there are still many open questions regarding retrograde transport mechanisms that allow for the efficient recycling of sorting receptors in the endomembrane

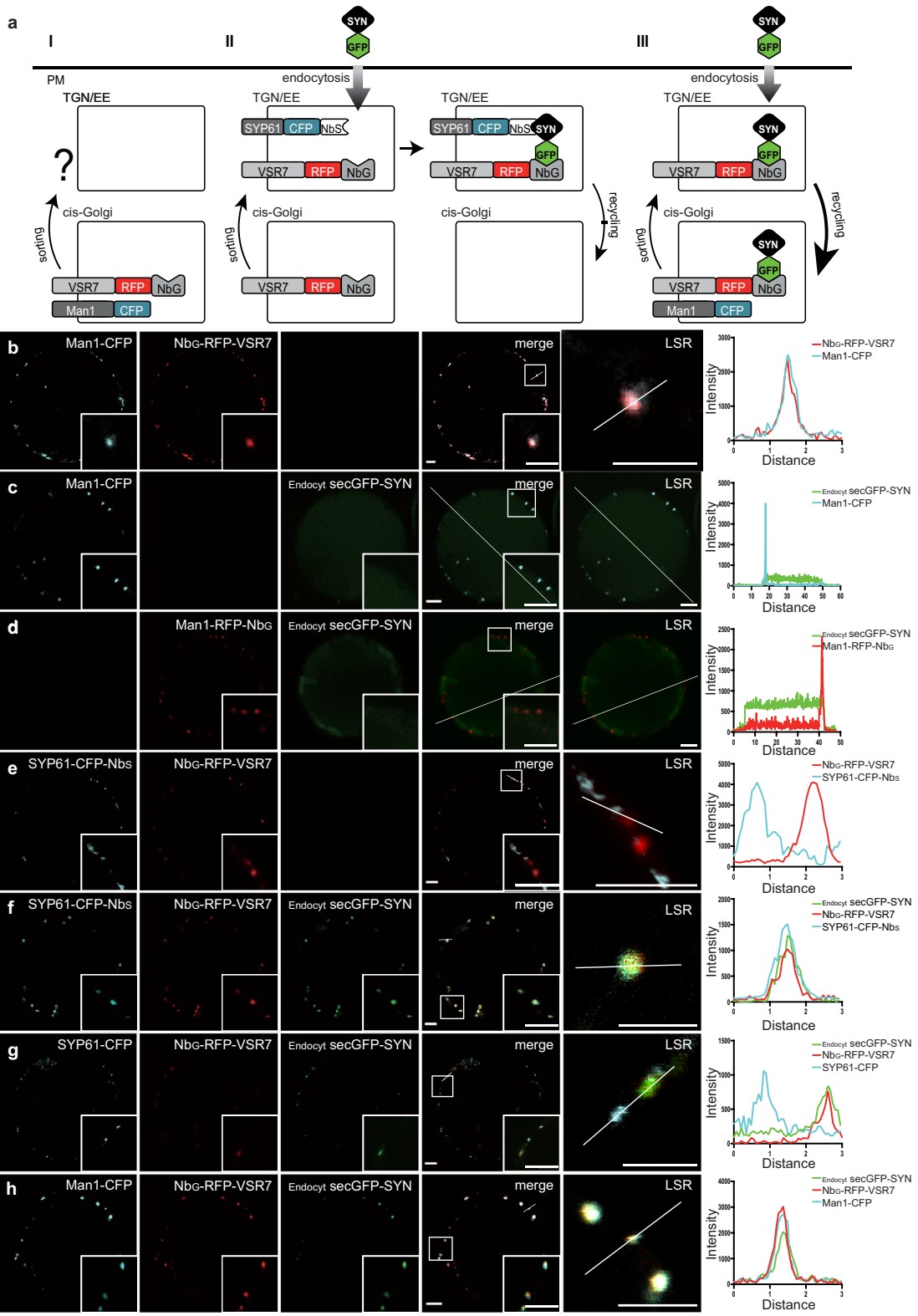

system of plants, which will be exciting research projects for the future.

## Methods

### Plant materials

*Nicotiana tabacum* L. SR1 was grown on Murashige and Skoog's medium supplemented with 2% (w/v) sucrose, 0.5 g L⁻¹ MES, and 0.8% (w/v) Agar at pH 5.7 in 16/8 h light-dark cycles at 22 °C.

### Protoplast isolation and transfection

Electrotransformation-competent tobacco protoplasts were isolated and transfected as previously described[53]. In short, about 2.5 million protoplasts in a total volume of 600 μL electro-transfection-buffer were electro-transfected with 1-10 μg plasmid DNA per construct, using a 160 V single square wave pulse for 10 ms (Gene Pulser Xcell™, Bio-Rad Laboratories ltd., Shanghai). After transfection, each sample was supplemented with 2 ml incubation buffer and

**Fig. 4 | The cis-Golgi-localizing VSR7 cycles between the cis-Golgi and the TGN/EE. a** Strategies and molecular tools for tracing the cis-Golgi-localizing VSR7 to reveal **I** the VSR7's downstream trafficking from the cis-Golgi to the TGN/EE, **II** its arrival in the TGN/EE, and **III** its TGN/EE transit and subsequent retrograde recycling to the cis-Golgi. **b** Coexpression of $Nb_G$-RFP-VSR7 (red) with Man1-CFP (blue) reveals the cis-Golgi localization of the $Nb_G$-tagged receptor. **c** Endocytic uptake of the protoplast-secreted dual-epitope linker secGFP-SYN (green) by protoplasts expressing Man1-CFP results in vacuolar delivery of the endocytosed molecule, but not in colocalization with the cis-Golgi marker. **d** Endocytic uptake of the protoplast-secreted dual-epitope linker secGFP-SYN (green) by protoplasts expressing Man1-RFP-$Nb_G$ (red) results in vacuolar delivery of the endocytosed molecule, but not in colocalization with the nanobody-tagged cis-Golgi marker (blue). **e** Coexpression of $Nb_G$-RFP-VSR7 (red) with SYP61-CFP-$Nb_S$ (blue) yields no overlap of signals from the cis-Golgi-localizing $Nb_G$-tagged receptor and the $Nb_S$-tagged TGN/EE marker. **f** Endocytic uptake of secGFP-SYN by $Nb_G$-RFP-VSR7 (red)

and SYP61-CFP-$Nb_S$ (blue)-expressing protoplasts results in the colocalization of all three fluorescence signals, demonstrating the arrival and subsequent trapping of the VSR7 in the TGN/EE (compare to a II). **g** Endocytic uptake of the secGFP-SYN (green) by cells coexpressing $Nb_G$-RFP-VSR7 (red) with the TGN/EE marker SYP61-CFP (blue) that lacks the $Nb_S$ causes posttranslational labeling of the transiting $Nb_G$-RFP-VSR7 in the TGN/EE before recycling (compare to a III). **h** Endocytic uptake of the secGFP-SYN (green) by cells coexpressing $Nb_G$-RFP-VSR7 (red) with the cis-Golgi marker Man1-CFP (blue) results in colocalization of the three signals demonstrating the TGN/EE transit and recycling to the cis-Golgi (Compare to a III). The colocalization/distribution of fluorescent proteins are visualized by overlapping/non-overlapping peaks in the line intensity plots. Insets are magnifications of the region indicated by a white square in the merged image. LSR, line scan region of the line intensity analysis. Scale bars = 5 μm. Experiments **a**−**i** were repeated at least twice with similar results; representative images are shown. Source data for **b**−**h** are provided as a Source Data file.

---

incubated for 16–24 h, if not indicated otherwise, at 25 °C in the dark.

### Genetic constructs
DNA manipulations were performed according to established procedures, using pGreenII-based vectors and *Escherichia coli* MC1061. All VSR constructs are based on *At*VSR7 (GenBank accession No. NM_001203848). The coding sequences of the GFP/α-synuclein-binding nanobodies were synthesized according to the *Arabidopsis* codon usage and were described previously[24,27]. All constructs used in this study are given in Supplementary Table 1.

### Confocal microscopy and image analysis
Image acquisition was performed using a confocal laser scanning microscope (Nikon A1 plus, Nikon, Japan) with a 40 × 1.15 NA water immersion objective. The BFP2, CFP, GFP, and RFP-containing fusion proteins were excited at λ 405 nm, 455 nm, 488 nm, and 561 nm, and emission was recorded in the range of 425–475 nm, 425–475 nm, 500–550 nm, and 570–620 nm, respectively. Pinholes were adjusted to 1 Airy unit for each wavelength. Post-acquisition image processing and analysis were performed using the software ImageJ (v.1.51, https://imagej.nih.gov/ij/index.html) and GraphPad Prism 8.0 (https://www.graphpad.com/scientific-software/prism).

### Statistics & reproducibility
The sample size for FLIM analysis was estimated based on previously achieved effect sizes in our lab: Effect size f (ANOVA) for the data shown in Fig. 1e was 1.24; with our desired error values (=0.001, (1-β) = 0.95) and 5 different groups of samples, this computes to a minimum of 35 total samples or 7 samples per group. Effect size f (ANOVA) for the data shown in Fig. 2e was 0.96; with our desired error values (=0.001, (1-β) = 0.95) and first 4 different groups of samples in the absence of the drug, the other 4 different groups of samples in the presence of the drug, this computes to a minimum of 17 total samples or 5 samples per group. The calculation was performed using G*Power Version 3.1.9.2. We increased this number to 10 samples per group to accommodate for possible slightly weaker effect sizes or data point distributions, which would necessitate alternative non-parametric tests.

For quantification of signal colocalization, the linear Pearson's correlation coefficient ($r_P$) and nonlinear Spearman's rank correlation coefficient ($r_S$) of fluorescent signals were calculated. The calculations were performed using ImageJ software with the PSC colocalization plug-in, and threshold levels were set to 10. For statistics, correlation coefficients of 10 individually analyzed cells per experiment were considered and are given as average values with SD. Statistical significance was calculated using ANOVA, followed by Student's t-test.

All experiments were repeated with similar results. The number of repetitions of each experiment is given in the respective figure legend.

### Data exclusion statement
Protoplast that were not completely turgescent or exhibited any signs of damage and protoplasts that did not express the respective fluorescent proteins were excluded from image acquisition. For FLIM analysis regions of interest (ROIs) were chosen in a way, that all fluorescent signals, which did not obviously originate from Chlorophyll autofluorescence, were used to calculate average lifetimes.

### Randomization statement
All protoplasts for an experiments were derived from a single pool, allocation into different transformations samples was performed by pipetting ~3*10^6 protoplasts at once. Thus individual cells were completely randomly distributed to the transformation samples.

### Blinding statement
Blinding was not performed during experiments and outcome assessment. Due to characteristic morphology, distribution, and movement of fluorescently tagged compartmental marker proteins in the cells, the investigators are able to deduct which sample they are observing.

### Fluorescence lifetime imaging microscopy (FLIM)
Experimental setup, data acquisition, and analysis were performed as previously described[54]. FLIM recordings were performed using a Nikon A1R confocal laser scanning microscope (Nikon, Japan) equipped with a PicoHarp 300 time-correlated single-photon counting (TCSPC) module and a PDL800-D picosecond diode laser driver (PicoQuant GmbH, Berlin, Germany). The energy donor GFP (GFP-CNX) was excited at λ 485-nm with 40-MHz pulse frequency. Emission was recorded at λ 482/35 nm until a count of at least 400 photons was reached in the brightest pixel. Per sample, 10-15 cells were recorded. FLIM data were analyzed using SymphoTime64 v2.0 (PicoQuant GmbH, Berlin, Germany). Values result from the analysis of selected regions of the cells using the software's "region of interest" (ROI) tools, allowing for the elimination of background noise. To calculate the fluorescence lifetimes of the donor, TCSPC histograms were reconvoluted with an instrumental response function (IRF) and fitted against a multi-exponential ($n = 2$) decay model. Only fittings with Chi-squared values between 0.9 and 1.5 were considered.

The calculated lifetimes of the respective cells are presented as box plots with whiskers showing all data points from min to max, with the box showing the interquartile range of the middle 50%, and the median is indicated by a line. Statistical significance was calculated as given above.

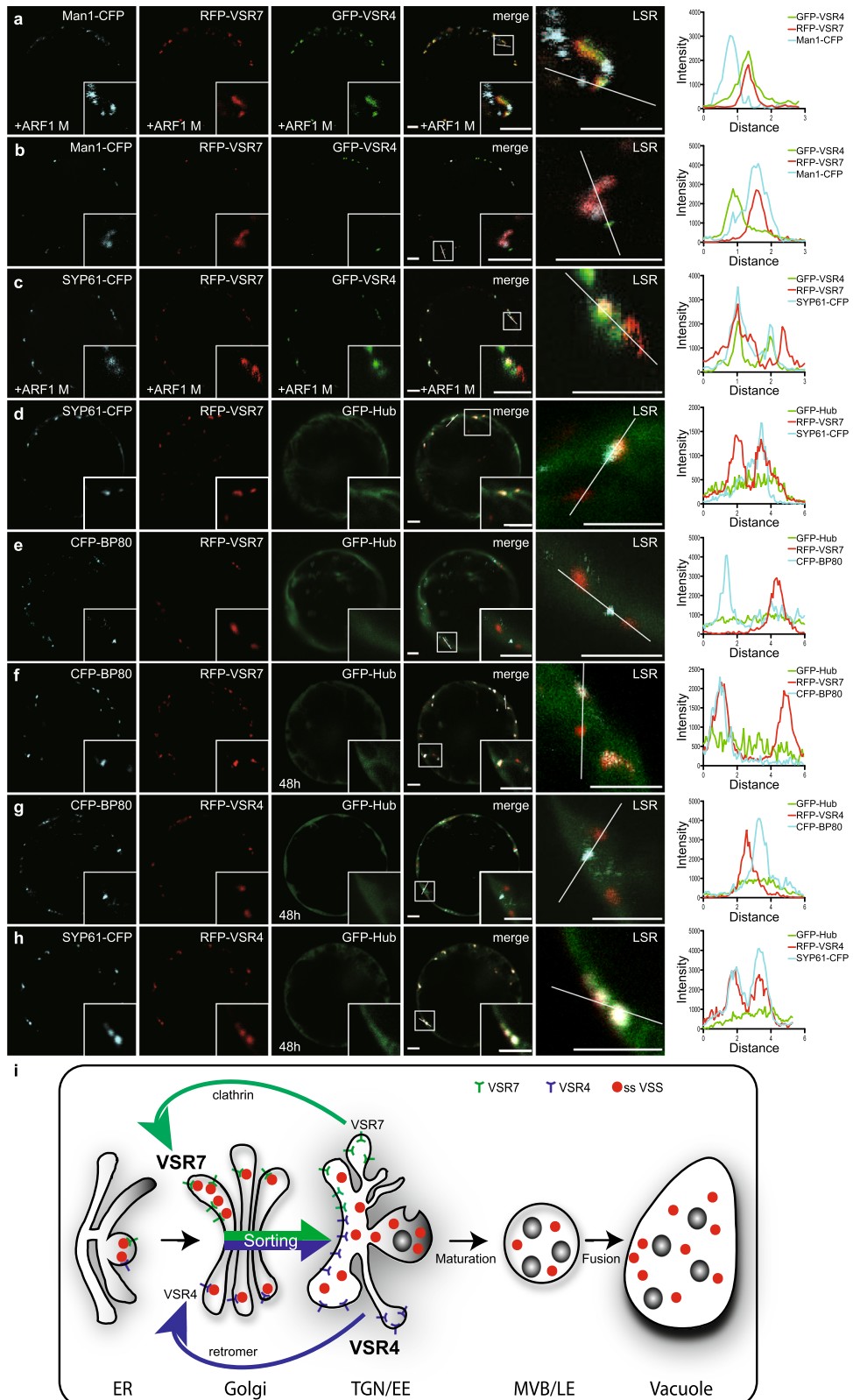

### Cell extraction and immunodetection

Extraction of cells and analysis by SDS–PAGE/WB was performed with minor modifications as previously reported[55]. All processed samples were mixed with an equal amount of 2× Xtreme loading dye (900 µL of sample buffer (0.1% (w/v) bromophenol blue, 5 mM EDTA, 200 mM Tris·HCl, pH 8.8, 1 M sucrose), supplemented with 300 µL 10% (w/v) SDS and 20 µl of 1 M DTT) and denatured for 5 min at 95 °C. The antibody used was a rat monoclonal anti-HA–Peroxidase antibody (Roche 12013819001, 1:5,000).

### Phylogenetic analysis

The amino acid sequences of the members of the VSR protein families from the given species were retrieved from the National Center for Biotechnology Information (https://www.ncbi.nlm.nih.gov) and

**Fig. 5 | Clathrin-dependent TGN/EE to *cis*-Golgi recycling of VSR7.**
**a** Coexpression of RFP-VSR7 (red), GFP-VSR4 (green), and Man1-CFP (blue) with the GTP-locked dominant-negative ARF1 mutant ARF1M, triggers the separation of VSR7 and Man1-CFP signals and causes colocalization of RFP-VSR7 with the TGN/EE-localizing GFP-VSR4, compared to **b** controls without ARF1M coexpression.
**c** Coexpression of RFP-VSR7 (red), GFP-VSR4 (green), and SYP61-CFP (blue) with ARF1M triggers the colocalization of VSR7 signals with the VSR4 and SYP61-CFP signals. **d** Coexpression of the fluorescent, dominant-negative mutant of the clathrin heavy-chain, GFP-Hub (green), with the RFP-VSR7 (red) and SYP61-CFP (blue) triggers the overlap of receptor and TGN/EE marker signals, indicating perturbed receptor recycling. **e, f** Coexpression of the GFP-Hub (green) with the RFP-VSR7 (red) and the MVB/LE marker CFP-BP80 (blue) for **e** 16 h expression and **f** 48 h expression, respectively, reveal the downstream leakage of the VSR7 from the TGN/EE toward the MVB/LE. **g** Coexpression of the GFP-Hub (green) with RFP-VSR4 (red) and CFP-BP80 (blue) for 48 h doesn't cause any downstream leakage of the VSR4

toward the MVB/LE. **h** Coexpression of the GFP-Hub (green) with the RFP-VSR4 (red) and SYP61-CFP (blue) for 48 h doesn't change the TGN/EE localization of the VSR4. Insets are magnifications of the region indicated by a white square in the merged image. LSR, line scan region of the line intensity analysis. **a-h** Scale bars = 5 μm. **i** Receptor-mediated vacuolar sorting in plants. After synthesis and folding in the ER, new sorting receptors can bind amino acid-encoded vacuolar sorting signals of soluble vacuolar proteins and receptor-ligand complexes are exported to the *cis*-Golgi. Anterograde transport from the *cis*-Golgi to the TGN/EE occurs via cisternal maturation and peeling off-based development of the *trans*-Golgi cisterna into a Golgi-independent TGN/EE. The development-accompanying biochemical changes of the compartmental lumen cause the release of the ligands, and the receptors undergo vesicle-mediated recycling to another *cis*-Golgi cisterna for the next round of transport, while released ligands reach the vacuole via the MVB/LE route. Experiments **a–h** were repeated at least twice with similar results; representative images are shown. Source data for **a–h** are provided as a Source Data file.

aligned using ClustalW (https://www.ebi.ac.uk/clustalw). The aligned VSR protein sequences were further processed using Molecular Evolutionary Genetics Analysis (MEGA) version 11 (https://www.megasoftware.net/) to generate Neighbor-Joining trees under 1000 replicates of bootstraps iterations.

### Reporting summary
Further information on research design is available in the Nature Portfolio Reporting Summary linked to this article.

## Data availability
All data supporting the findings of this study are available within the paper and its Supplementary Information. Source data are provided with this paper.

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

## Acknowledgements

This work was supported by the National Natural Science Foundation of China No. 31970185 P.P, and No. 32270742.P.P, and the Key Laboratory of Molecular Design for Plant Cell Factory of Guangdong Higher Education Institutes (2019KSYS006), and the Shenzhen Science and Technology Program (No. KQTD20190929173906742). We thank the staff at SUSTech Cryo-EM Center for their assistance. We thank Dr. Yelin Zhou and Dr. Xibin Lu for their technical support on FRET–FLIM.

## Author contributions

X. S. and H. X. performed experiments and data acquisition. X.S. and P.P. evaluated the data and wrote the manuscript, and P. P. conceived the study.

## Competing interests

The authors declare no competing financial interests.
