## [Peer Review File · Nature Communications]

Reviewers' Comments:

Reviewer #1:

Remarks to the Author:

The mechanisms and trafficking routes of plant vacuolar sorting receptor (VSR) proteins are incompletely understood. This manuscript focuses on the ligand specificity and trafficking route of the VSR7 receptor, using nanobodies and epitope tags as molecular markers. Some of the approaches are sophisticated and novel. The basic conclusions are: (a) VSR7 binds ligands in the cis-Golgi but cannot bind ligands in the trans-Golgi network/early endosome (TGN/EE), presumably due to the lower pH, and (b) VSR7 resides at steady-state in the cis-Golgi but travels to the TGN/EE and then recycles in a clathrin-dependent manner.

The data presented in support of these conclusions are generally persuasive. In vivo experiments offer strong evidence that VSR7 binds to the vacuolar reporter Aleu-RFP but not to the vacuolar reporter RFP-AFVY, indicating that VSR7 has specificity for a subset of vacuolar sorting signals. Further experiments indicate that the VSR7-ligand interaction can occur in the ER and cis-Golgi but not in the TGN/EE. The analysis of the role of clathrin is a bit convoluted, so those conclusions are more provisional, but clathrin likely has a role in VSR7 trafficking. Overall, this study is technically impressive and it makes a noteworthy contribution to the field.

My major concern is the interpretation of the pathway followed by VSR7. The authors propose that VSR7 binds ligands in the cis-Golgi and then releases them in the TGN/EE, but the rationale for this pathway is puzzling if the Golgi operates by cisternal maturation. Why would a receptor be needed to carry proteins forward to the TGN/EE when maturation will do the job? A relevant review by Robinson and Neuhaus (<https://pubmed.ncbi.nlm.nih.gov/27262127/>) should be cited, because it puts forth a model in which vacuolar sorting receptors segregate cargoes into specialized domains during transport through the Golgi. To this reader, the new data on VSR7 fit that model nicely. The authors should offer their perspective to ensure that their results contribute to our understanding of how VSR7 contributes to vacuolar protein sorting.

Minor points:

- 1) p. 4: "RFP-VSR4 (Suppl. Fig. 3)" should read "RFP-VSR4 (Suppl. Fig. 4)".
- 2) p. 7: "did not colocalize with the TGN/MVB marker" should presumably read "did not colocalize with the TGN/EE marker".
- 3) There is evidence for clathrin adaptor-dependent recycling from the TGN to the cis/medial Golgi in yeast and mammalian cells (<https://pubmed.ncbi.nlm.nih.gov/35429729/>). That pathway may be related to the proposed recycling pathway for VSR7.

Reviewer #2:

Remarks to the Author:

The authors used transient assay in tobacco protoplasts and the NB-based sensor and showed that VSR7 binds to ALEU in the early secretory pathway and dissociates from it at the TGN/EE. They also showed that VSR7 localizes to the cis-Golgi in a steady state. Using a system that can trap the Nb-sensor at the TGN/EE, the authors also showed that VSR7 is recycled after reaching the TGN. The fact that VSR7 accumulates in the TGN upon overexpression of ARF1DN or Hub indicated that CCV is required for recycling.

The trap experiment presented in Fig. 4 is interesting, but the overall data is not well presented and the conclusions are not convincing.

major

The problem is that there is no quantitative data for colocalization/non-colocalization.

The lower right corners of pictures in Fig. 2-5 and supplemental data appear to be enlarged images, but where in the image is enlarged is unclear. Many of the dots in the enlarged images are not found in the whole cell image, perhaps because they are taken from another cell or rotated.

Also, it is difficult to locate the line for a line plot. If a magnified image is to be included, it would be better to take it from the presented image.

The experiment in Fig.4 is interesting, but the control experiment is lacking, and from the current results, it cannot be determined where the binding took place. For example, authors should show that RFP does not go to the TGN, or GFP does not come to the cis-Golgi, using the cis-Golgi resident SYP3-RFP-NbG.

minor

I think the reference to the numbers of VSRs and RMRs in page 2 refers to Arabidopsis and does not apply to higher plants in general.

I don't understand the rationale for choosing the genes authors used for the phylogenetic analysis in Suppl 1. Also, because tobacco cells are used in this experiment, it would be better to show that there are Class III VSRs in tobacco as well.

For VSR5/6, which also belong to Class III, it has been reported that OX of VSR5 in the *vsr1vsr4* double mutant promotes AALP secretion (Lee et al., 2013 <https://doi.org/10.1104/pp.112.210914>). Whether VSR5/6 and VSR7 have different functions in the same Class III or it is due to distinct experimental systems should be verified by additional experiments, if possible.

Bottom of page 3, RFP-AFVY is not accumulated in the ER, so it should not be surprising that FRET does not occur.

As for the TGN localization of VSR7, VSR7 has been detected in proteome analyses of the TGN using VHAa1 and SYP61 (Groen et al., 2014 <https://doi.org/10.1021/pr4008464>, Drakakaki et al., 2011 [doi:10.1038/cr.2011.129](https://doi.org/10.1038/cr.2011.129)). These should be cited and discussed in the manuscript.

There is no data to exclude the possibility that VSR7 is recycled from the MVB, therefore the title should be an overstatement. More careful discussion would be needed.

In the Discussion, the authors state that VSR4 and VSR7 should have the same velocity of anterograde transport in the Golgi. What is the rationale for this?

Reviewer #3:

Remarks to the Author:

In this manuscript Shao et al investigates the trafficking routes of VSR7, an understudied vacuolar sorting receptor and suggest that VSR7 bind cargoes in the cis-Golgi to then release them in downstream TGN/EE compartment. In the paper they use a previously published strategy based on the interaction between GFP and a GFP nanobody to test the trafficking of VSR7. The sensor is able to detect VSR7 in the cis-Golgi. They use FLIM to test interaction of different vacuolar cargoes and show that VSR7 is responsible for the trafficking of an Aleu-RFP cargo. They localize the full length receptor to the cis-Golgi and develop an assay to show that VSR7 recycles back to the Golgi from TGN/EE. Additionally, they show that VSR7 trafficking is clathrin dependent by using a clathrin mutant.

In conclusion they suggest intra Golgi transport of a sub-set cargoes in plants may be a receptor-dependent process. This is a very interesting model and a novel concept across species as intra-Golgi transport in mammals (debated) and in yeast is thought to be driven by cisternal maturation. In my opinion this is a very bold claim which in this case is unfortunately not supported by the data. It would need careful evaluation and quantification of the imaging data before it can be published. I cannot suggest this work and I feel like I am unable to evaluate this work until proper quantification is carried out.

The authors must carry out quantification and statistics on independent experiments. With crops with one single dot structure, it is impossible to evaluate how sound the data is... For example, in Figure 2B I do see colocalization between ManI and Aleu which makes me worry that the authors may have biasedly picked crops.

I understand that the author claim that cargo binds to the receptor in the cis-Golgi based on their FLIM interaction data (of which I am not an expert and cannot evaluate the soundness...). However, I fail to see data in support of release of cargo in the TGN/EE.

I also recommend that the authors move away from using non-color blind friendly red and green to show their microscopy images.

A point-by-point response to the reviewers' comments

Reviewer #1

The mechanisms and trafficking routes of plant vacuolar sorting receptor (VSR) proteins are incompletely understood. This manuscript focuses on the ligand specificity and trafficking route of the VSR7 receptor, using nanobodies and epitope tags as molecular markers. Some of the approaches are sophisticated and novel. The basic conclusions are: (a) VSR7 binds ligands in the cis-Golgi but cannot bind ligands in the trans-Golgi network/early endosome (TGN/EE), presumably due to the lower pH, and (b) VSR7 resides at steady-state in the cis-Golgi but travels to the TGN/EE and then recycles in a clathrin-dependent manner.

The data presented in support of these conclusions are generally persuasive. *In vivo* experiments offer strong evidence that VSR7 binds to the vacuolar reporter Aleu-RFP but not to the vacuolar reporter RFP-AFVY, indicating that VSR7 has specificity for a subset of vacuolar sorting signals. Further experiments indicate that the VSR7-ligand interaction can occur in the ER and cis-Golgi but not in the TGN/EE. The analysis of the role of clathrin is a bit convoluted, so those conclusions are more provisional, but clathrin likely has a role in VSR7 trafficking. Overall, this study is technically impressive and it makes a noteworthy contribution to the field.

Response:

We thank the reviewer for the very positive feedback regarding the scientific significance of this work and the appreciation of the quality of our work.

My major concern is the interpretation of the pathway followed by VSR7. The authors propose that VSR7 binds ligands in the cis-Golgi and then releases them in the TGN/EE, but the rationale for this pathway is puzzling if the Golgi operates by cisternal maturation. Why would a receptor be needed to carry proteins forward to the TGN/EE when maturation will do the job? A relevant review by Robinson and Neuhaus (<https://pubmed.ncbi.nlm.nih.gov/27262127/>) should be cited, because it puts forth a model in which vacuolar sorting receptors segregate cargoes into specialized domains during transport through the Golgi. To this reader, the new data on VSR7 fit that model nicely. The authors should offer their perspective to ensure that their results contribute to our understanding of how VSR7 contributes to vacuolar protein sorting.

Response:

Here, the reviewer raises an exciting topic, and we strongly agree with the reviewer's view. Our data regarding the receptor-ligand interactions and receptor trafficking of the VSRs VSR7 and VSR4 point to a sorting mechanism of the VSRs, that differs from the classical concept of receptor-mediated transport of soluble ligands in which ligand transport is assumed to occur via vesicle shuttles that connect spatially distinct compartments, with one providing binding conditions, while the other provides release conditions. Regarding VSR-mediated transport in plants, it seems as if the VSR-mediated transport of ligands does not occur between spatially distinct but temporarily distinct compartments, with the target compartment emerging as the result of a maturation-based process. In this concept, it seems as if the sorting function of a VSR is to bind the ligand at neutral pH in the *cis*-Golgi cisternae and immobilize it throughout the cisternae's maturation into a VHA-a1-acidified Golgi-independent TGN with EE functionality to prevent its secretory loss. This view is, indeed, in full agreement with the proposed concept from Robinson and Neuhaus, 2016, *J. Exp. Bot.* 67, 4435-4449. We modified the discussion regarding the VSR-based transport mechanism and added this citation to the manuscript.

Minor points:

1) p. 4: "RFP-VSR4 (Suppl. Fig. 3)" should read "RFP-VSR4 (Suppl. Fig. 4)".

Response:

We apologize for the confusion; we have corrected the mistake.

2) p. 7: "did not colocalize with the TGN/MVB marker" should presumably read "did not colocalize with the TGN/EE marker".

Response:

We apologize for the confusion; we have corrected the mistake.

3) There is evidence for clathrin adaptor-dependent recycling from the TGN to the cis/medial Golgi in yeast and mammalian cells (<https://pubmed.ncbi.nlm.nih.gov/35429729/>). That pathway may be related to the proposed recycling pathway for VSR7.

Response:

We appreciate this information and we have included this aspect in the discussion.

Reviewer #2

The authors used transient assay in tobacco protoplasts and the NB-based sensor and showed that VSR7 binds to ALEU in the early secretory pathway and dissociates from it at the TGN/EE. They also showed that VSR7 localizes to the cis-Golgi in a steady state. Using a system that can trap the Nb-sensor at the TGN/EE, the authors also showed that VSR7 is recycled after reaching the TGN. The fact that VSR7 accumulates in the TGN upon overexpression of ARF1DN or Hub indicated that CCV is required for recycling.

The trap experiment presented in Fig. 4 is interesting, but the overall data is not well presented and the conclusions are not convincing.

Response:

We are sorry if the presentation of the data has caused confusion. In this revised manuscript, we have substantially revised all Figures according to the suggestions below.

Major

The problem is that there is no quantitative data for colocalization/non-colocalization.

Response:

We agree with the reviewer, and we have therefore included the quantification of the colocalization of the respective receptors and marker proteins for the analyses of receptor localization and receptor trafficking that are presented in Fig. 3, Fig. 4, and Fig. 5 and Supplementary. Fig. 8 by calculating the linear Pearson's correlation coefficient (r_P) and nonlinear Spearman's rank correlation coefficient (r_S) of fluorescent signals using the software ImageJ with the PSC colocalization plugin. We furthermore performed a statistical analysis of the respective PSCs for a better appreciation in general and the localization changes observed in the trafficking analysis, in particular. We present the scatter plots together with the statistical analysis as the three new supplementary figures 5, 6, and 7 and added the details to the Methods section. In this regard, Supplementary. Fig. 5 confirms that GFP-VSR7 colocalizes with the *cis*-Golgi marker Man1-RFP but not with the TGN/EE-localizing VSR4, the TGN/EE marker SYP61-RFP, the MVB/LE marker RFP-BP80, or the *trans*-Golgi marker ST-RFP; Supplementary. Fig. 6 confirms that the endocytosed dual-epitope linker secGFP-SYN, the labeling agent of the trafficking analysis, does not reach the *cis*-Golgi on its own and also underlines the labeling efficiency of the nanobody-epitope interaction-triggered posttranslational labeling of the Nb_G-RFP-VSR7 and the SYP61-CFP-Nbs; Supplementary. Fig. 7 confirms that compared to control cells where the RFP-VSR7 colocalizes with the *cis*-Golgi marker Man1-CFP but not with the TGN/EE-localizing GFP-VSR4 or the TGN/EE marker SYP61-CFP, the expression of the ARF1 mutant causes a significant reduction of the colocalization between the RFP-VSR7 and the *cis*-Golgi marker Man1-CFP and a significant

increase of the colocalization with the GFP-VSR4. Likewise, the coexpression of the GFP-Hub causes a significant rise in the colocalization of the RFP-VSR7 with the TGN/EE marker SYP61-CFP. Prolonged expression of the GFP-Hub for 48 hours causes the RFP-VSR7 to colocalize with the MVB/LE marker CFP-BP80, while RFP-VSR4 colocalizes with the TGN/EE marker SYP61-CFP, but not with the MVB/LE marker CFP-BP80.

The lower right corners of pictures in Fig. 2-5 and supplemental data appear to be enlarged images, but where in the image is enlarged is unclear. Many of the dots in the enlarged images are not found in the whole cell image, perhaps because they are taken from another cell or rotated. Also, it is difficult to locate the line for a line plot. If a magnified image is to be included, it would be better to take it from the presented image.

Response:

We fully agree with the view regarding the insets and the line scan region (LSR) indications. For the previous version of the figures, we first imaged the respective cell and then took a second image of a section of that cell at a higher magnification, which we presented as an inset to visualize the respective pattern of the fluorescent signals. During the time this took, the intracellular compartments moved. Therefore, it was not possible to obtain identical signal patterns in both images, and we could not indicate the respective regions in the image of the cell. In this revised version, we simplified the situation by using an enlarged region of the respective cell as an inset and framed this region in the image of the respective cell with a white square. We also performed the line intensity analysis of a region in the inset and presented this region as an additional enlargement, labeled line scan region, LSR, in the respective panels. These revisions have been done to Fig. 2, 3, 4, and 5 and Supplementary Fig. 3, 4, and 8.

The experiment in Fig. 4 is interesting, but the control experiment is lacking, and from the current results, it cannot be determined where the binding took place. For example, authors should show that RFP does not go to the TGN, or GFP does not come to the *cis*-Golgi, using the *cis*-Golgi resident SYP3-RFP-NbG.

Response:

We thank the reviewer for appreciating the experimental strategy to identify TGN/EE transit and recycling of the *cis*-Golgi localizing VSR7. The reviewer has requested an additional control to rule out that the in Fig. 4 observed nanobody-epitope interaction-triggered posttranslational labeling of the at steady-state in the *cis*-Golgi-localizing Nb_G-RFP-VSR7 with the endocytosed secGFP-SYN occurred due to TGN/EE to *cis*-Golgi upstream trafficking of the labeling agent, rather than downstream trafficking of the Nb_G-RFP-VSR7 to the TGN/EE and its subsequent recycling to the *cis*-Golgi, as claimed. For this, the reviewer suggested exposing the Nb_G with a red fluorescent *cis*-Golgi-localizing protein for trapping endocytosed secGFP-SYN molecules upon arrival in the *cis*-Golgi. We appreciated this suggestion, but for the sake of consistency, we decided to keep using *cis*-Golgi-localizing marker/anchor constructs based on Man1 throughout the manuscript rather than introducing a new SYP3-based *cis*-Golgi marker at this point, as was suggested by the reviewer, and generated the red fluorescent Nb_G-tagged *cis*-Golgi-localizing anchor Man1-RFP-Nb_G for the requested control experiments. The endocytic uptake assays with secGFP-SYN and Man1-RFP-Nb_G-expressing cells revealed secGFP-SYN signals in the vacuole, but no colocalization between the endocytosed secGFP-SYN and the *cis*-Golgi-localizing Man1-RFP-Nb_G was observed, demonstrating that the endocytosed secGFP-SYN does not reach the *cis*-Golgi upon endocytic uptake by default and that the posttranslational labeling of the *cis*-Golgi-localizing Nb_G-RFP-VSR7 with the secGFP-SYN must have occurred due to the downstream trafficking of the receptor to the TGN/EE. This new result nicely supplements the presented control in which we used Man1-CFP expressing cells in endocytic uptake assays with secGFP-SYN (Fig. 4c) and is therefore inserted as a new panel in the revised Fig 4 as Fig. 4d, with the quantification being presented in the new Supplementary Fig. 6. This is also in agreement with the published data showing that the endocytic uptake of such soluble fluorescent

proteins leads via the TGN/EE and the MVB/LE to the vacuole, independent of sorting receptors (Künzl et al., 2016, Nature Plants 2, 16017).

However, we were confused regarding the above request to show that "RFP does not go to the TGN." In the context of the trafficking analysis of VSR7 (Fig. 4), RFP is translationally fused to the Nb_G-tagged full-length VSR7, Nb_G-RFP-VSR7, and the results show that this molecule does reach/transit the TGN/EE (Fig. 4f). Therefore, we decided to omit this suggestion. Together with the new control, it seems now justified to claim that the colocalization of the *cis*-Golgi marker Man1-CFP with the posttranslationally endocytosed secGFP-SYN-labeled Nb_G-RFP-VSR7 (Fig. 4h) demonstrates that Nb_G-RFP-VSR7 did transit the TGN/EE, acquired the labeling due to the nanobody-epitope interaction in the TGN/EE and recycled back to the *cis*-Golgi afterward.

Minor

I think the reference to the numbers of VSRs and RMRs in page 2 refers to Arabidopsis and does not apply to higher plants in general.

Response: We thank the reviewer for spotting this. We have clarified in the manuscript that the numbers given for VSRs and RMRs refer to *Arabidopsis*.

I don't understand the rationale for choosing the genes authors used for the phylogenetic analysis in Suppl 1. Also, because tobacco cells are used in this experiment, it would be better to show that there are Class III VSRs in tobacco as well.

Response:

We decided to show the phylogenetic relationship of members of the VSR family from different species to emphasize the classification of the VSR family. As suggested, we have now also included the three published VSR sequences from *Nicotiana tabacum* and identified the VSRs NtVSR1 as class I, NtVSR3 as class II, and NtVSR6 as class III VSRs.

For VSR5/6, which also belong to Class III, it has been reported that OX of VSR5 in the *vsr1vsr4* double mutant promotes AALP secretion (Lee et al., 2013 <https://doi.org/10.1104/pp.112.210914>). Whether VSR5/6 and VSR7 have different functions in the same Class III or it is due to distinct experimental systems should be verified by additional experiments, if possible.

Response:

This is an exciting question, but it isn't easy to answer and is even more challenging to investigate. The function of sorting receptors is complex and includes reversible cargo interaction and bidirectional transport of the receptor. Therefore, it isn't easy to judge "function" in general. Lee et al. (2013, Plant Physiol. 161, 121-133) present a set of sophisticated and exceptionally well-executed experiments. They used vacuolar trafficking-impaired *vsr1vsr4* mutant protoplasts for coexpressing increasing amounts of HA-tagged VSRs (either VSR1, VSR4, or VSR5 but not the VSR6 or VSR7) with a constant amount of the soluble vacuolar cargo AALP:GFP, and analyzed the amount of reporter in cell extracts and culture medium by SDS-PAGE/WB to assess transport and to draw conclusions about the involvement of the respective VSR in the transport of the cargo into the vacuole. In the case of the VSR5, increased expression levels of the receptor resulted in an increased amount of "non-sorted" vacuolar cargo that reached the culture medium (Lee et al., 2013, Fig. 6). While it is evident that the expressed VSR5 did not recover the mutant phenotype, it is not clear why the increased amount of the receptor gradually enhanced the *vsr1vsr4* phenotype. At first sight, an increase in ligand-binding-competent receptors is expected to reduce/prevent secretory loss, while an increase in the number of ligand-binding-incompetent receptors is expected not to alter the number of missorted cargo molecules at all. However, the above experiment (Lee et al., 2013, Fig. 6) revealed a VSR5 dosage-dependent increase in the cargo sorting defect. The most plausible way to explain such an enhancement of the sorting deficiency of the mutant line is that the expression of the VSR5 also interfered with the transport of the sorting-relevant endogenous receptors, which reduced the already low sorting efficiency of the mutant even further. One possibility for such influences could be

the competition between the endogenous receptors and the expressed receptors for coat proteins or other transport machinery components, either for the anterograde or retrograde direction, that could cause the transport to collapse, and receptors would continue to accumulate. At such conditions, accumulating even a ligand-binding competent VSR in a compartment that does not provide ligand-binding conditions would not reduce the secretory loss of cargo in the mutant.

In an attempt to tackle the question regarding similar or different functions of class III VSRs, we have generated a VSR5-based sensor system and assessed the ligand-binding ability of the VSR5 via FRET/FLIM, similar to what was performed for the analysis of the VSR7 (shown in Fig. 1e). The confidential Fig. C1 below shows that the ssVSS-carrying ligand Aleu-RFP also triggers the reduction of the VSR5 sensor's fluorescence lifetime, while the ctVSS-carrying RFP-AFVY and the non-ligand RFP-HDEL does not. This suggests that VSR7 and VSR5 exhibit the same ligand binding competence for the ssVSS-carrying ligand Aleu-RFP *in vivo*.

Confidential Figure C1: CLSM analysis of VSR5 sensor-ligand interaction in tobacco mesophyll protoplasts. **(a)** FLIM assessment of VSR5 sensor-ligand interactions, showing I) Energy donor (GFP-CNX) only, or the assembled VSR5 sensor GFP-CNX/LBD5-NbG with II) Aleu-RFP, III) RFP-AFVY, IV) RFP-HDEL or V) the GFP-RFP dual-color sensor GFP-CNX/LBD5-RFP-NbG. Fluorescence lifetimes are given as GFP-CNX fluorescence lifetime in ns as box plots. Significance was calculated using one-way ANOVA, followed by Tukey's HSD test (**** $P < 0.0001$ compared with every other group; NS, not significant, $N=10-15$ cells). Images in the rows below the chart show false color fluorescence lifetime analysis (FLIM), and the respective localization analysis of the proteins (green channel, red channel, and merge). Scale bars = 5 μm . **(b)** Control of expression of the VSR5 sensing unit. SDS-PAGE/WB analysis showing the expression of the HA-tagged sensing unit LBD5-NbG in coexpression with the ER anchor GFP-CNX and the respective RFP-fusion proteins used for the FLIM analysis shown in a, using anti-HA antibodies.

Regarding the experiment presented by Lee et al., 2013, increased secretion of AALP:GFP by a ssVSS-binding-competent VSR5 in the *vsr1vsr4* mutant would argue for a transport competition situation between the receptors, as discussed above. Therefore it would be crucial to know the steady-state

distribution of the endogenous VSRs of the *vsr1vsr4* line, which are relevant for the AALP:GFP sorting, and the localization of the expressed VSRs regarding their binding/release locations. However, VSR7 and VSR5 do not colocalize. Therefore, it must be assumed that their transport mechanisms differ. However, concluding receptor functionality between the class III VSRs VSR5, VSR6, and VSR7 is a very complicated issue that requires complex analysis of their ligand spectra and mapping of their intracellular trafficking routes, which exceeds the scope of this manuscript. At this stage, we can not provide sufficient data for a conclusive answer to this question, and we, therefore, prefer omitting this topic from the manuscript.

Bottom of page 3, RFP-AFVY is not accumulated in the ER, so it should not be surprising that FRET does not occur.

Response:

We are sorry about this mistake. We were surprised by the lack of colocalization between the sensor and the ctVSS carrying cargo RFP-AFVY. We have clarified this in the manuscript.

As for the TGN localization of VSR7, VSR7 has been detected in proteome analyses of the TGN using VHAa1 and SYP61 (Groen et al., 2014 <https://doi.org/10.1021/pr4008464>, Drakakaki et al., 2011 doi:10.1038/cr.2011.129). These should be cited and discussed in the manuscript.

Response:

We thank the reviewer for this information, and we have added the notion that the VSR7 has been identified in fractions of immunoprecipitated SYP61-CFP or vacuolar-type H⁺-ATPase subunit a1 (VHA-a1)-GFP compartments in proteomic analyses, and we have added the respective references Drakakaki et al., 2012, Cell Res. 22, 413-424 and Groen et al., 2014, J. Proteome Res. 13, 763-776, accordingly.

There is no data to exclude the possibility that VSR7 is recycled from the MVB, therefore the title should be an overstatement.

Response:

With all due respect, we do not agree with this view. Even though recycling from the MVB/LE in plants is controversial (Robinson and Neuhaus, 2016, J. Exp. Bot. 67, 4435-4449.), it seems that this case does not apply to the recycling of the VSR7 in particular. Our newly added quantification and statistical analysis of the VSR7 trafficking analysis show that the inhibition of clathrin-mediated trafficking by the expression of the GFP-Hub alters the distribution of the *cis*-Golgi-localizing VSR7 and causes its accumulation at the TGN/EE first, while the accumulation at the MVB/LE occurs only after a prolonged-expression for 48 h. If the demonstrated clathrin-dependent recycling of the VSR7 occurred from the MVB/LE rather than the TGN/EE, the GFP-Hub would have caused the accumulation of the VSR7 at the MVB/LE first, which is not the case.

Regarding putative recycling of VSRs from the MVB/LE, it is noteworthy to say that - at least to our knowledge - no experimental evidence has been provided that demonstrates that clathrin-coated vesicles transport VSRs with soluble vacuolar cargo to the MVB/LE, or that VSRs depart from an MVB/LE in a clathrin-coated vesicle for retrograde recycling to the *cis*-Golgi (Robinson and Neuhaus, 2016, J. Exp. Bot. 67, 4435-4449.).

It, therefore, seems justified to claim that the VSR7 recycles from the TGN/EE to the *cis*-Golgi.

More careful Discussion would be needed. In the Discussion, the authors state that VSR4 and VSR7 should have the same velocity of anterograde transport in the Golgi. What is the rationale for this?

Response:

We are sorry that this has caused confusion. In the Discussion, we did not intend to state that "VSR4 and VSR7 should have the same velocity of anterograde transport in the Golgi" in general. We referred to a model situation in which the anterograde transport of membrane proteins across the Golgi stack would occur via a maturation-based cisternal progression mode that employs the Golgi-associated COPI transport vesicles only for the retrograde recycling of Golgi-resident proteins but not

for selective anterograde transport of cargo proteins. For this situation, we assumed that the transport of membrane proteins would occur at the same speed. We have clarified this in the manuscript.

Reviewer #3 (Remarks to the Author):

In this manuscript Shao et al., investigates the trafficking routes of VSR7, an understudied vacuolar sorting receptor and suggest that VSR7 bind cargoes in the cis-Golgi to then release them in downstream TGN/EE compartment. In the paper they use a previously published strategy based on the interaction between GFP and a GFP nanobody to test the trafficking of VSR7. The sensor is able to detect VSR7 in the cis-Golgi. They use FLIM to test interaction of different vacuolar cargoes and show that VSR7 is responsible for the trafficking of an Aleu-RFP cargo. They localize the full length receptor to the cis-Golgi and develop an assay to show that VSR7 recycles back to the Golgi from TGN/EE. Additionally, they show that VSR7 trafficking is clathrin dependent by using a clathrin mutant.

In conclusion they suggest intra Golgi transport of a sub-set cargoes in plants may be a receptor-dependent process. This is a very interesting model and a novel concept across species as intra-Golgi transport in mammals (debated) and in yeast is thought to be driven by cisternal maturation. In my opinion this is a very bold claim which in this case is unfortunately not supported by the data.

Response:

With all due respect, we do not claim that the *intra*-Golgi transport of soluble vacuolar proteins across the stack is a receptor-dependent process. However, we agree with the reviewer that our data would not support such a claim.

In transportation, the term "receptor-dependent" refers to a transport situation that does not occur without the receptor. This does particularly not apply to the anterograde intra-Golgi transport of soluble proteins across the stack. The Golgi stack is a central part of the secretory pathway that is taken by soluble secretory proteins, which don't possess sorting signals, by default. Likewise, soluble Golgi-transiting proteins that carry sorting signals for post-Golgi locations, like the vacuole, are also secreted in mutants that lack the respective receptors. Therefore, intra-Golgi transport across the stack can't be termed receptor-dependent. However, post-Golgi transport of soluble proteins to locations other than the plasma membrane requires sorting signals and, consequently, the interaction with sorting receptors, as is the case for the VSR-mediated sorting of soluble vacuolar proteins. In this regard, we demonstrate that the VSR7 binds specifically to the ssVSS-carrying cargo Aleu-RFP *in vivo*. We show that the VSR7 binds Aleu-RFP in the ER, the *cis*-Golgi, but not in the TGN/EE. Our localization analysis shows that the VSR7, in contrast to other VSRS, localizes at steady state conditions at the *cis*-Golgi, trafficks to the TGN/EE for the release of ligands, and we provide evidence for its clathrin-dependent retrograde recycling to the *cis*-Golgi.

It would need careful evaluation and quantification of the imaging data before it can be published. I cannot suggest this work and I feel like I am unable to evaluate this work until proper quantification is carried out. The authors must carry out quantification and statistics on independent experiments. With crops with one single dot structure, it is impossible to evaluate how sound the data is... For example, in Figure 2B I do see colocalization between ManI and Aleu which makes me worry that the authors may have biasedly picked crops.

Response:

We agree with the reviewer and apologize for leaving the impression of having biasedly picked crops. The lack of quantification of the CLSM localization data has also been raised by reviewer #2, and we have replied to this in detail above. In short, we have quantified the colocalization of the respective receptors and marker proteins for the analyses of receptor localization and trafficking presented in Fig. 3, Fig. 4, and Fig. 5 and Supplementary. Fig. 8 by calculating Pearson's correlation coefficient (r_P) and nonlinear Spearman's rank correlation coefficient (r_S) of the respective fluorescent signals and

performed a statistical analysis of the PSCs. We present these data in three new supplementary figures, Supplementary Fig. 5, Supplementary Fig. 6, and Supplementary Fig. 7. We appreciate this suggestion and acknowledge that the added data have significantly strengthened our claims.

I understand that the author claim that cargo binds to the receptor in the cis-Golgi based on their FLIM interaction data (of which I am not an expert and cannot evaluate the soundness...). However, I fail to see data in support of release of cargo in the TGN/EE.

Response:

We appreciate this comment. Visualizing the compartment-specific ligand release *in vivo* is most challenging, particularly in the TGN/EE, because neither the VSR7 nor the identified ligand Aleu-RFP - or any other soluble vacuolar protein localizes at the TGN/EE at steady-state condition. We concluded that the VSR7 releases the ligand in the TGN/EE since it is the first compartment of the vacuolar transport pathway, that, in contrast to the ER, cis- and *trans*-Golgi, provided no colocalization-based evidence of an occurring interaction between the VSR7 sensor and the identified ligand Aleu-RFP (Fig. 2a-d). In the protoplasts, the red fluorescent protein-tagged vacuolar cargo causes strong red vacuolar background signals and, additionally, accumulates in the MVB/LE (red punctae) at steady-state conditions, while the green fluorescent sensor is localized in TGNs/EEs (green punctae), that in cortical view appear on top of the strong vacuolar background. In this situation, selecting regions of interest for quantification is challenging. To solve this problem, we implemented FRET/FLIM to quantify the VSR7 sensor-ligand interaction in the TGN, which we included as a new panel in Fig. 2, Fig. 2e). In this analysis, we included the drug concanamycin A (Conc A), an established inhibitor of the TGN/EE-localizing vacuolar-type H⁺-ATPase subunit $\alpha 1$ (VHA- $\alpha 1$) that alters the pH of the TGN and thus blocks the post-TGN/EE transport to the vacuole. The new Figure 2e confirms that the ligand Aleu-RFP does not interact with the VSR7 sensor in the TGN/EE at control conditions but shows strong interaction in the presence of the drug. This is the first demonstration of a pH-dependent VSR-ligand interaction *in vivo* and provides now strong evidence for the TGN/EE as the ligand-release compartment of the VSR7.

I also recommend that the authors move away from using non-color blind friendly red and green to show their microscopy images.

Response:

We are aware that this is an important issue, and we envisaged the presentation of the microscopy data following guidelines for color blindness provided by Wong, 2011, Nat. Methods 8, 441; This, however, did not yield satisfactory results regarding the recognition and judgment of the varying combinations of overlapping signals in the three-channel imaging analysis in the context of the compartment-specific signal pattern of the protoplasts.

Reviewers' Comments:

Reviewer #1:

Remarks to the Author:

The authors have done a satisfactory job of addressing my concerns. I support publication of the revised manuscript.

Reviewer #2:

Remarks to the Author:

The authors addressed all of my concerns in the revised manuscript, and now I support publication of this manuscript, after slight modification of the text. In the newly added part on page 4-5, authors mention that concanamycin A is a specific inhibitor of the TGN/EE-localizing vacuolar-type H⁺-ATPase subunit $\alpha 1$ (VHA- $\alpha 1$). This is incorrect; ConCA is an inhibitor of V-ATPase in the TGN and vacuole, whose effect is not restricted to a specific subunit of the V-ATPase at the TGN.

A point-by-point response to the reviewers' comments

Reviewer #1 (Remarks to the Author):

The authors have done a satisfactory job of addressing my concerns. I support publication of the revised manuscript.

Response:

We are pleased to hear that we have satisfactorily addressed all comments and concerns, and we thank the reviewer for the efforts and inspirations that helped us to improve our manuscript.

Reviewer #2 (Remarks to the Author):

The authors addressed all of my concerns in the revised manuscript, and now I support publication of this manuscript, after slight modification of the text.

In the newly added part on page 4-5, authors mention that concanamycin A is a specific inhibitor of the TGN/EE-localizing vacuolar-type H⁺-ATPase subunit a1 (VHA-a1). This is incorrect; ConcA is an inhibitor of V-ATPase in the TGN and vacuole, whose effect is not restricted to a specific subunit of the V-ATPase at the TGN.

Response:

We are glad that our revisions have addressed all of the concerns. We fully agree with the reviewer's comment that the effect of the drug Conc A is not restricted to a specific subunit of the V-ATPase at the TGN. We have rephrased the respective sentence in the manuscript. We also thank this reviewer for the efforts and inspirations that helped us to improve our manuscript.